# Longitudinal Study of the Effects of *Flammulina velutipes* Stipe Wastes on the Cecal Microbiota of Laying Hens

Jiali Wei,[a] Huanwei Xiao,[a] Yingbo Wei,[a] Ivan Stève Nguepi Tsopmejio,[a] Chang Sun,[a] Haoyuan Wu,[a] Zhouyu Jin,[a] Hui Song[a,b,c]

[a]School of Life Science, Jilin Agricultural University, Changchun, Jilin, People's Republic of China
[b]Engineering Research Center of Chinese Ministry of Education for Bioreactor and Pharmaceutical Development, Changchun, Jilin, People's Republic of China
[c]Engineering Research Center of Chinese Ministry of Education for Edible and Medicinal Fungi, Changchun, Jilin, People's Republic of China

**ABSTRACT** Because antibiotics have been phased out of use in poultry feed, measures to improve intestinal health have been sought. Dietary fiber may be beneficial to intestinal health by modulating gut microbial composition, but the exact changes it induces remain unclear. In this study, we evaluated the effect of *Flammulina velutipes* stipe wastes (FVW) on the cecal microbiotas of laying chickens at ages spanning birth to 490 days. Using clonal sequencing and 16S rRNA high-throughput sequencing, we showed that FVW improved the microbial diversity when they under fluctuated. The evolvement of the microbiota enhanced the physiological development of laying hens. Supplementation of FVW enriched the relative abundance of *Sutterella*, *Ruminiclostridium*, *Synergistes*, *Anaerostipes*, and *Rikenellaceae*, strengthened the positive connection between *Firmicutes* and *Bacteroidetes*, and increased the concentration of short-chain fatty acids (SCFAs) in early life. FVW maintains gut microbiota homeostasis by regulating Th1, Th2, and Th17 balance and secretory IgA (S-IgA) level. In conclusion, we showed that FVW induces microbial changes that are potentially beneficial for intestinal immunity.

**IMPORTANCE** Dietary fiber is popularly used in poultry farming to improve host health and metabolism. Microbial composition is known to be influenced by dietary fiber use, although the exact FVW-induced changes remain unclear. This study provided a first comparison of the effects of FVW and the most commonly used antibiotic growth promoter (flavomycin) on the cecal microbiotas of laying hens from birth to 490 days of age. We found that supplementation with FVW altered cecal microbial composition, thereby affecting the correlation network between members of the microbiota, and subsequently affecting the intestinal immune homeostasis.

**KEYWORDS** dietary fiber, *Flammulina velutipes*, flavomycin, microbiota, laying hens

The microbial communities inhabiting the gastrointestinal tract play an important role in nutrient digestion, energy utilization, and immune system regulation (1). These regulatory effects are mediated by the complex microbial interactions and metabolites generated by the members of the microbial community (2, 3). The intestinal microbiota possesses genes encoding active enzymes of carbohydrates, which can decompose dietary fiber which is not digested by the host and produce short-chain fatty acids (SCFAs), mainly acetate, propionate, and butyrate (4). SCFAs affect host energy utilization. First, SCFAs, especially butyrate, are the energy substrates for colonic cells (5), and second, propionate is a substrate for the gluconeogenesis that can induce intestinal gluconeogenesis, signaling through the central nervous system to protect the host from diet-induced obesity and associated glucose intolerance (6). Third, acetate may help to improve metabolic health by increasing energy expenditure through whole-body fat browning (7). SCFAs can also act as signaling molecules to regulate host immunity. The combination of propionate and butyrate effectively inhibits

Address correspondence to Hui Song, songhuisk@jlau.edu.cn.

The authors declare no conflict of interest.

the lipopolysaccharide (LPS)-induced inflammatory response of regulatory T cells (Treg cells) and reduces the production of inflammatory cytokines such as interleukin 6 (IL-6) and IL-12 (8).

A healthy gut microbial state, characterized by a high diversity of microorganisms, improves the functional diversity as well as microbe-microbe and host-microbe interactions. It is also called an equilibrium state or steady state (9, 10). In contrast, microbial imbalance can induce inflammatory responses mediated by Th1, Th2, and Th17 cells (11, 12). The Th cells activated by intestinal epithelial cells induce B cells to produce and secrete antibodies on the surface of the intestinal mucosa, mainly secreted immunoglobulin A (IgA). The secretory immunoglobulin A (S-IgA) may promote the retention of beneficial members of the intestinal flora and the removal of opportunistic pathogens through different binding methods (13).

Antibiotic growth promoters (AGPs) have played a decisive role in animal husbandry for more than half a century (14). Among them, flavomycin (synonyms: bambermycin, monomycin, flavophospholipol) is a typical glycolipid phosphate antibiotic that acts on Gram-positive bacteria and mainly plays a role in promoting the growth performance of chickens (15, 16). However, the overuse of antibiotics has led to a rise in antimicrobial resistance. In response to this threat to public health, the European Union introduced a ban on the use of antibiotics as growth promoters in 2006 (17). Therefore, this crutch of the poultry industry must be replaced.

In recent years, a growing number of scientific studies have shown positive effects of dietary fiber on chicken health and productivity (18–20). Feeding experiments were mainly carried out with insoluble fiber sources that arise as by-products during industrial production, such as oat hulls, sunflower hulls, soybean hulls, wheat bran, and wood shavings (21). *Flammulina velutipes* is one of the most popular edible fungi and is rich in biological nutrients (carbohydrate, dietary fiber, glycoproteins, polyphenols, etc.) (22, 23). The annual output of *F. velutipes* in China exceeded 2.5 million tons from 2013 to 2019 (24); meanwhile, large amounts of the by-product *F. velutipes* stipe wastes (FVW) have also been produced. Previous studies have found that dietary FVW supplementation has no reverse regulation effect on the growth of laying hens but can increase antibody titers, enhance immune responses, promote calcium deposition in eggshells, and improve antioxidant capacity in serum and egg yolk (25–28).

The phylogenetic composition of the microbiota commonly found in different gut segments of broilers has been well characterized (29), whereas the literature describing this characteristic of laying hens is very limited. The long-term diet is strongly associated with the composition, activity, and dynamics of the gut microbiome, while short-term dietary changes are often not sufficient to elicit major changes in the ecosystem (30, 31). Laying hens live up to approximately 70 weeks before their laying rate decreases to about 65%. The previous studies reported the influence of microbial changes on outcome at different ages, preventing interstudy comparisons (32–34). Therefore, this study was designed to estimate the effects of FVW or flavomycin (FLA) on the cecal microbiota at different feeding stages by clonal sequencing and 16S rRNA gene analysis. The final purpose was to elucidate the regulatory effects of FVW on the microbiotas of laying hens. This study will provide a theoretical basis for using FVW as a prebiotic to maintain intestinal health of laying hens.

## RESULTS

**The diversity of the cecal microbiota in laying hens evolves with age and diet.** The impact of FVW on the cecal microbiota was assessed over a long study period, 490 days, with 450 laying hens divided into five diet supplementation groups (Fig. 1). Ninety chickens were randomly divided into 3 replicates in each group and received 5 ppm FLA, low (2%) FVW (LFVW), medium (4%) FVW (MFVW), high (6%) FVW (HFVW), or basic (unsupplemented) diet (BD). Nine chickens in each group were sacrificed and cecal contents were collected at day 7 (prestarter), day 28 (starter), day 70 (grower), day 112 (developer), and day 490 (finisher) to assess longitudinal microbial development.

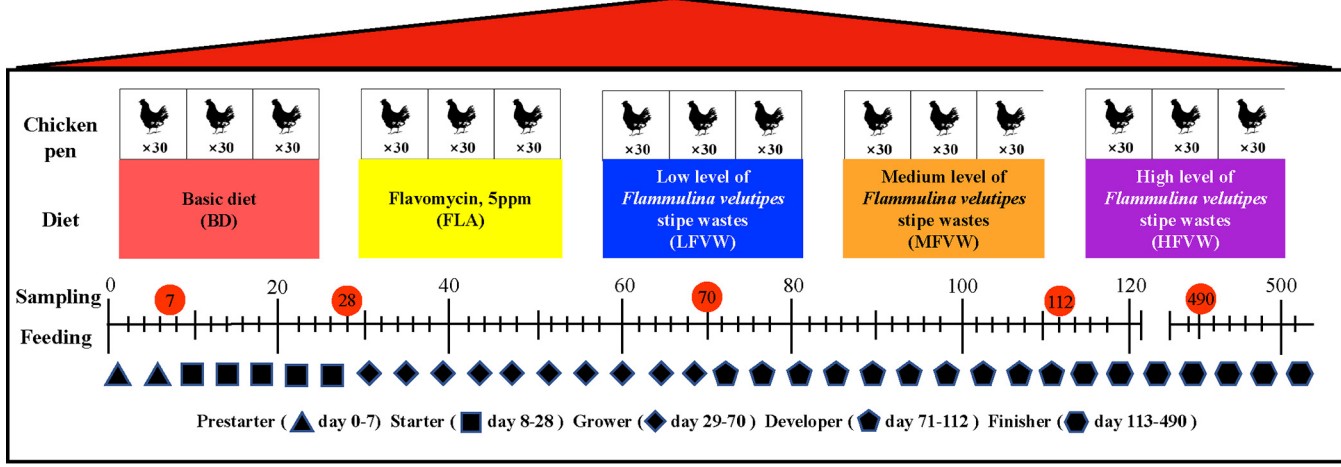

**FIG 1** Overview of feeding and sampling strategies. A total of 450 1-day-old ISA laying hens were randomly divided into five groups, as shown. Each group was divided into 3 replicates with 30 chickens per replicate. Feed for laying hens was formulated for prestarter (age 0 to 7 days), starter (age 8 to 28 days), grower (age 29 to 70 days), developer (age 71 to 112 days), and finisher (age 113 to 490 days) production phases. Samples were collected at the end of each phase, as indicated by time points marked with orange circles.

To investigate the evolution of the cecum microbiota over time, alpha diversities were compared at the end of the five phases (prestarter, starter, grower, developer, and finisher). In general, the richness, diversity, and evenness gradually increased over time, while the richness and diversity were lower in the starter phase ($P < 0.01$), and this might be due to the cascading perturbations of external factors (Fig. 2A to C). In the starter phase, richness and diversity were higher in the FVW and FLA groups than in the BD group ($P < 0.01$) (Fig. 2D and E; also, see Fig. S1A to C in the supplemental material). In the grower phase, the richness was higher in the LFVW and MFVW groups than the BD group ($P < 0.05$) (Fig. 2D). In the developer phase, the evenness was lower in the HFVW group than the FLA group ($P < 0.05$) (Fig. 2F). This might indicate that FVW could improve the diversity of the gut microbiota when there are fluctuations in the gut microbiota development.

The discriminant distribution between samples from different feeding phases were described by principal-component analysis (PCA). Analysis of similarities (ANOSIM) showed that flock development exerted a substantial influence on overall community variations ($P = 0.0001$) (Fig. 3A). In the starter, grower, and developer phases, samples from the MFVW and HFVW groups clustered separately from those of the BD and FLA groups (Fig. 3C to E), while in the finisher phase, the cecal microbiotas of laying hens tended to be homogenous ($P = 0.1567$) (Fig. 3F; Fig. S1D). Chickens receiving the BD and the FLA-supplemented diet harbored similar microbial profiles, as they clustered together in each of five phases (Fig. 3B to F). Thus, the increase in community diversity was accompanied by decreased heterogeneity of the cecal microbiota.

**Succession of dominant gut microbiotas in the cecum of laying hens.** The succession of dominant gut microbiotas in the cecum of laying hens was traced at the taxonomic levels of phylum, class, order, family, and genus (Fig. 4; Fig. S2). *Firmicutes* was the dominant phylum in the prestarter and starter phases, while *Firmicutes*, *Bacteroidetes*, and *Proteobacteria* were dominant in the last three phases (Fig. 4A). *Clostridiales* was the dominant order throughout the feeding phases of laying hens, with a relative abundance greater than 80% in the first two phases, while in the last three phases, *Bacteroidales* and *Clostridiales* were both dominant (Fig. 4B). At the family level, *Ruminococcaceae* were predominant in the prestarter phase. During the starter phase, *Lachnospiraceae* transitioned to the dominant family. During the grower phase, *Bacteroidaceae* replaced *Lachnospiraceae* as the dominant family, while the predominance of *Bacteroidaceae* was diluted by *Eubacteriaceae* and *Pseudomonadaceae* in developer phase. At the finisher phase, *Prevotellaceae*, *Eubacteriaceae*, *Rikenellaceae*, and *Bacteroidaceae* became the dominant families (Fig. 4C). At the genus level, *Ruminococcus*, *Bariatricus*, *Intestinimonas*,

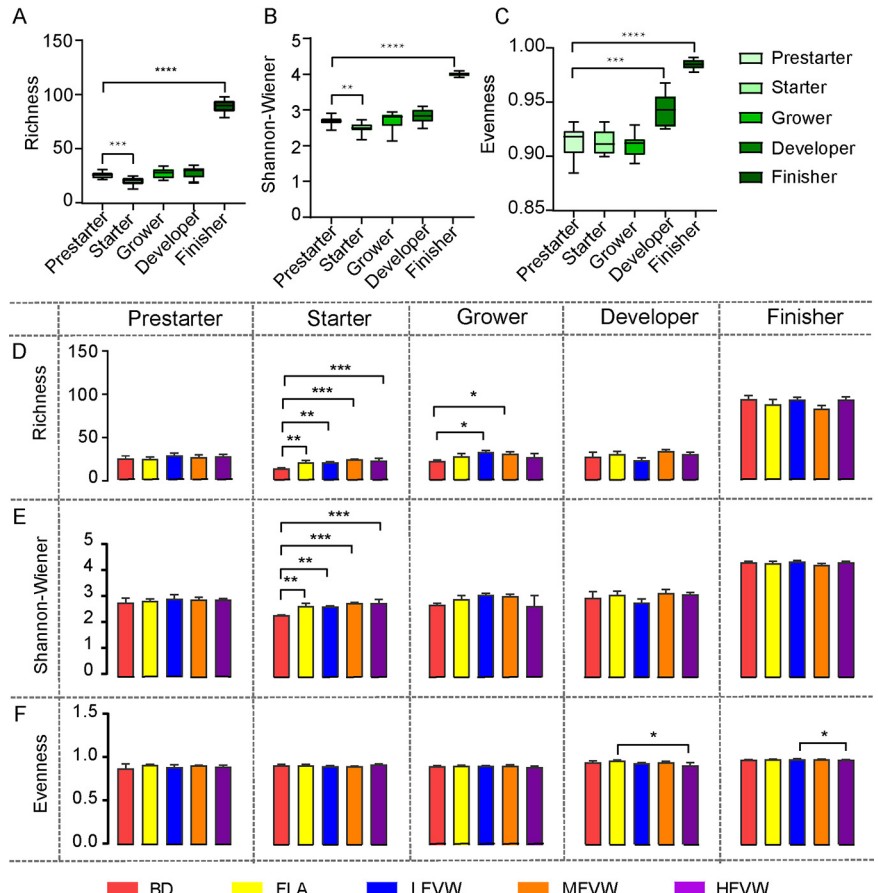

**FIG 2** The distribution of alpha diversity represents changes in the diversity of gut microbiota in the cecum of laying hens by age and supplementation. (A to C) Richness (A), Shannon-Wiener (B), and evenness (C) indices indicate that the overall diversity increases with phases. (D to F) Richness (D), Shannon-Wiener (E), and evenness (F) indices indicate that the relative diversity changes by supplementation. There were statistically significant differences in community structure among different supplemental groups. *P* values were adjusted for multiple testing using one-way ANOVA. *, $P < 0.05$; **, $P < 0.01$; ***, $P < 0.001$; ****, $P < 0.0001$.

*Pseudoflavonifractor*, and *Lachnoclostridium* were predominant in the prestarter phase. *Lachnoclostridium*, *Blautia*, and *Roseburia* were dominant in the starter phase. *Bacteroides*, *Ruminiclostridium*, and *Barnesiella* were dominant in the grower phase, while *Pseudoflavonifractor*, *Bacteroides*, *Eubacterium*, and *Acinetobacter* were the most important in the developer phase. Finally, *Alistipes*, *Eubacterium*, and *Prevotella* were the dominant genera in the finisher phase (Fig. 4D). At the end of the feeding period (in the finisher phase), the linear discriminant analysis effect size (LEfSe) multilevel discriminant analysis of species differences indicated that compared to the BD group, the FLA group was differentially enriched in *Ruminococcaceae*, *Desulfovibrio*, *Peptococcus*, *Enterorhabdus*, and *Faecalicoccus* (Fig. 5A). In the LFVW group, *Sutterella* and *Ruminiclostridium* were differentially enriched (Fig. 5B), while *Synergistes* and *Anaerostipes* were enriched in the MFVW group (Fig. 5C) and *Rikenellaceae* and *Synergistes* in the HFVW group (Fig. 5D).

**FVW supplementation altered the co-occurrence network of the cecal microbiota.** The pattern of interbacterial interactions among the cecal microbial communities was analyzed by constructing the co-occurrence network of each group (based on Spearman correlation). The metacommunity cooccurrence networks of the BD, FLA, LFVW, MFVW, and HFVW groups comprised, respectively, 159 edges, 154 edges, 143 edges, 159 edges, and 152 edges, representing 30 interactive genera (Fig. 6). In the FLA group, the largest numbers of positive relationships between *Firmicutes* and *Firmicutes*, *Bacteroidetes*, and *Bacteroidetes* and of negative relationships between *Firmicutes* and

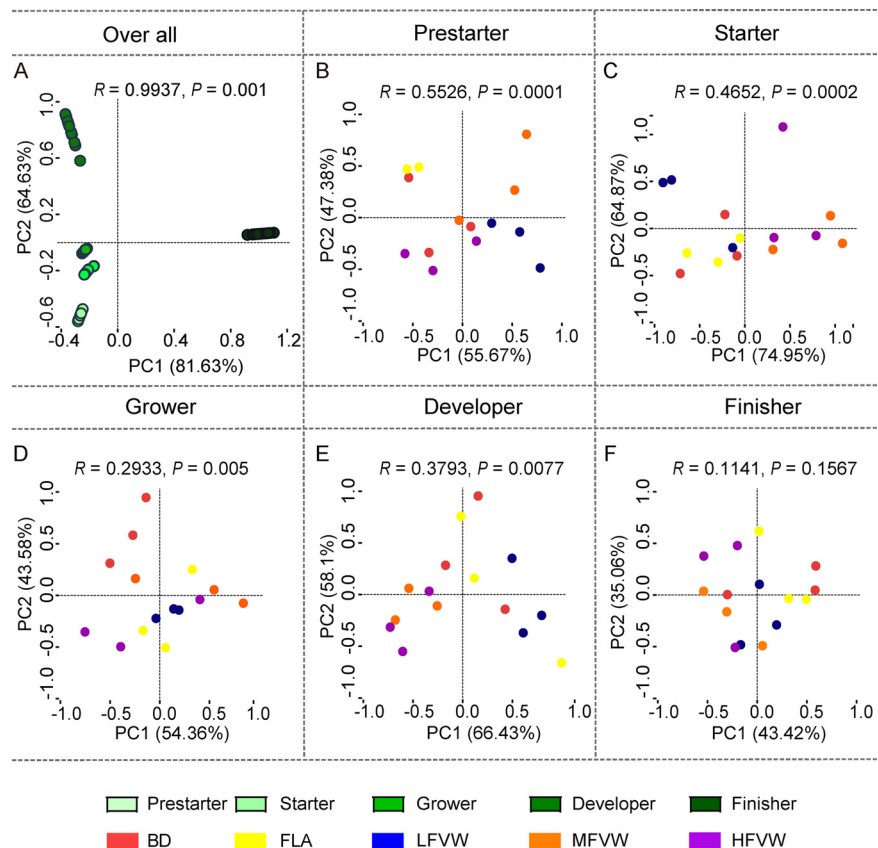

**FIG 3** Differences in microbial composition were demonstrated using PCA of Bray-Curtis dissimilarity in all phases (A) and in the prestarter (B), starter (C), grower (D), developer (E), and finisher (F) phases. Beta diversity relationships are summarized in two-dimensional scatterplots. Each point represents a sample, and distances between dots are representative of differences in microbiota compositions. There were statistically significant differences in community structure among different supplement groups (*R* > 0, *P* < 0.05).

*Bacteroidetes* were found. Meanwhile, the fewest negative correlations between *Firmicutes* and *Firmicutes*, *Bacteroidetes,* and *Bacteroidetes* and positive correlations between *Firmicutes* and *Bacteroidetes* were also found (Fig. 6B). The long-term supplementation of FVW changed the interaction between members of the gut microbiota, as shown by the strengthening positive correlation between *Firmicutes* and *Bacteroidetes* and the weakening positive correlation between *Firmicutes* and *Firmicutes*. Among them, MFVW supplementation caused the most significant changes in microbial interactions (Fig. 6A to E). These results might indicate that the supplementation of fiber-rich FVW affects the composition of carbon sources available to microorganisms and then changes the energy utilization among the intestinal microorganisms.

**FVW supplementation increased SCFA concentrations in the early life of laying hens.** The microbiota in the cecum ferments undigested carbohydrates by using its own glycohydrolytic activity to produce SCFAs (35). SCFAs play a role in regulating immunity by acting on epithelial and immune cells (36). Acetate, propionate, and butyrate were maintained at a stable level throughout the feeding phases of laying hens (Table 1). The FVW supplementation accelerated SCFA production in the early life (prestarter and starter phases) of laying hens, while FLA mainly promoted the production of SCFAs during the grower and developer phases (*P* < 0.05) (Table 1). In the prestarter phase, acetate levels were higher in the MFVW (9.05 ± 0.44 mmol/L) and HFVW (9.39 ± 0.33 mmol/L) groups than in the BD (7.26 ± 0.10 mmol/L) and FLA (6.62 ± 0.04 mmol/L) groups (*P* < 0.5). Broadly, the highest level of propionate was found in the LFVW group. In the starter phase, acetate, propionate, and butyrate were significantly increased in the FLA and FVW supplementation groups compared to the

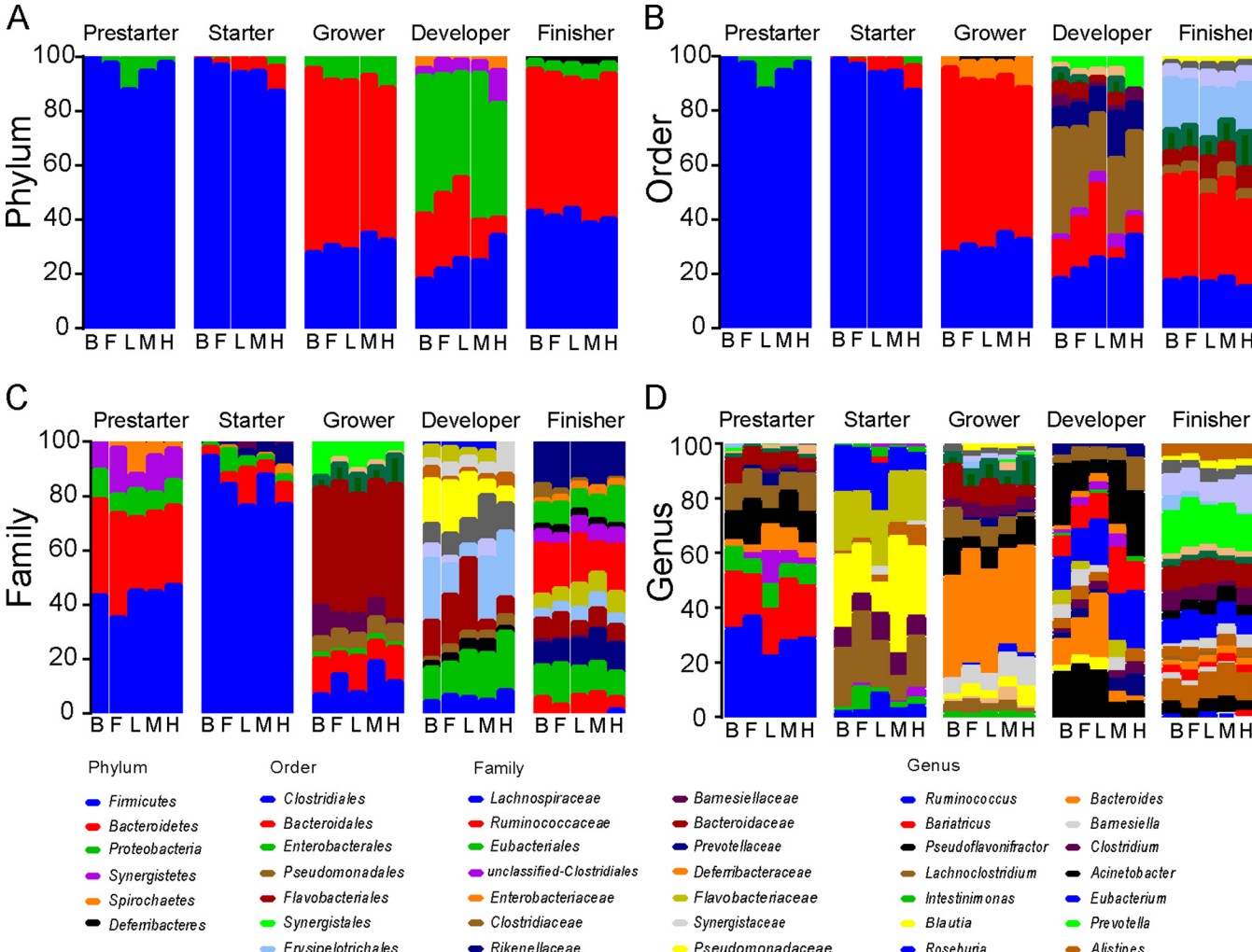

**FIG 4** Composition and succession of gut microbiota in laying hens. Stacked bar plots represent the diet-induced distortions in the main phylum (A), order (B), family (C), and genus (D). B, BD; F, FLA; L, LFVW; M, MFVW; H, HFVW.

BD group ($P < 0.05$). In the grower and developer phases, the FLA group showed the highest concentration of acetate and butyrate, while the HFVW group showed the lowest concentration of SCFAs in general ($P < 0.05$). In the finisher phase, SCFAs were significantly reduced in the FLA and FVW groups ($P < 0.05$) (Table 1).

**FVW regulated the homeostasis of intestinal mucosal immunity in laying hens.** The role of the microbiome is especially crucial in early life for the development of the immune system. A microbial imbalance could induce inflammatory responses mediated by Th1, Th2, and Th17 cells (12). Th1 and Th17 cells secrete proinflammatory cytokines, while Th2 cells secrete anti-inflammatory cytokines. The levels of the proinflammatory cytokines tumor necrosis factor alpha (TNF-$\alpha$) and IL-6 were significantly reduced in the FVW groups compared with those in the BD and FLA groups during the prestarter and starter phases ($P < 0.05$) (Fig. S3A to D). IL-2 is known to promote the differentiation of the anti-inflammatory cytokine IL-10 (37). The level of IL-2 significantly increased in the FVW compared to the BD and FLA groups ($P < 0.05$) (Fig. S3G and H). The levels of the anti-inflammatory cytokine IL-4 significantly increased in the MFVW and HFVW groups (Fig. S3I). FVW supplementation significantly increased the levels of S-IgA in the small intestinal mucosa compared to the BD and FLA groups ($P < 0.05$) (Fig. S3M and N). The levels of IL-6 and IL-2 were significantly increased in the FLA group compared to the BD group (Fig. S3C and H). The other cytokines did not change significantly between groups (Fig. S3). Therefore, FVW could regulate the

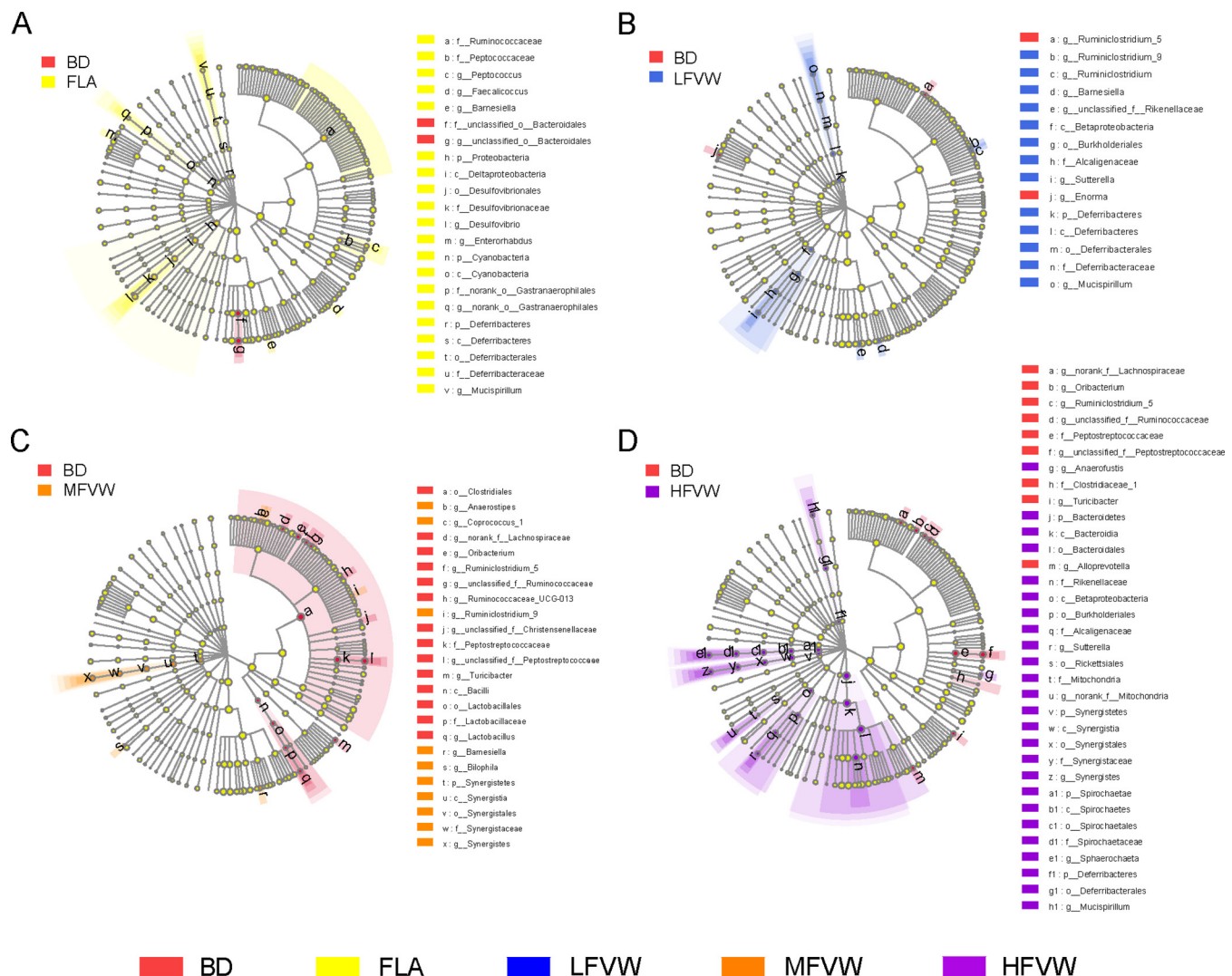

**FIG 5** Shifts in taxa of the main cecal microbiomes of laying hens. LEfSe multilevel discriminant analysis of species differences in the finisher phase between (A) BD and FLA, (B) BD and LFVW, (C) BD and MFVW, and (D) BD and HFVW.

dynamic balance of intestinal mucosal immunity in laying hens, which might be beneficial for the homeostasis of the commensal microbiota.

## DISCUSSION

The gut microbiota of laying hens plays a crucial role in host health and development. Dietary fiber could act as a prebiotic to regulate the intestinal health of chickens (38), but the effects of long-term dietary interventions on the gut microbiotas of laying hens were largely unknown. Therefore, our study aimed to evaluate the effects of fiber-rich *Flammulina velutipes* stipe waste (FVW) on the microbiota of laying hens, using FLA (flavomycin), an antibiotic growth promoter commonly used in laying hens, as a reference for 70 weeks. FVW induced beneficial changes in the cecal microbiota of laying hens. It changed the interaction network between bacteria and regulated the homeostasis of intestinal mucosal immunity. These results demonstrate that the long-term feeding of fiber-rich FVW may serve as a potential prebiotic alternative to the use of antibiotic growth promoters.

FVW increased the diversity of the gut microbiota in laying hens during starter phase. A higher diversity of gut microbiota is associated with a healthier physiological state (39). The factors influencing gut microbiota diversity include host factors and

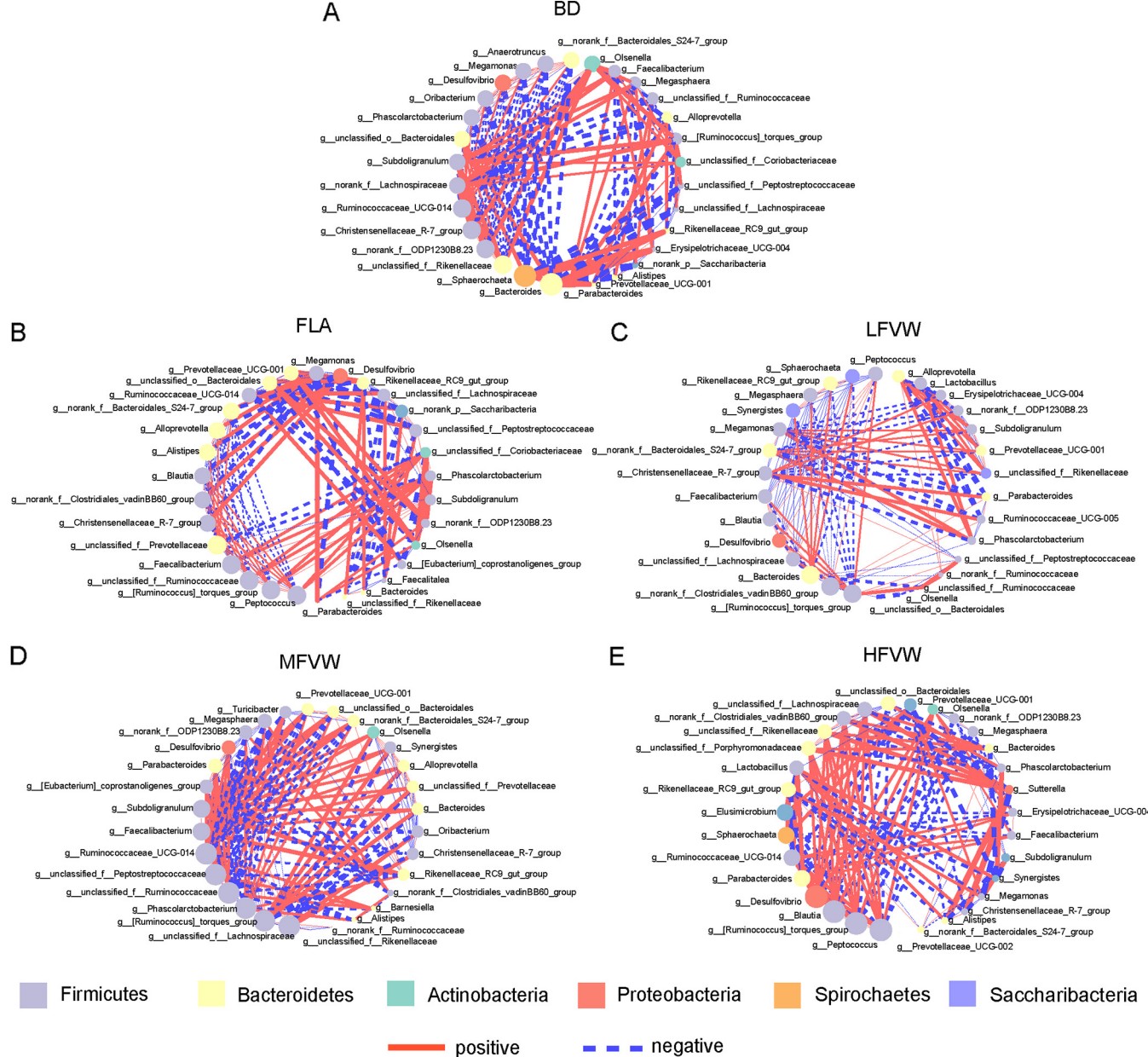

**FIG 6** Spearman's correlation network analyses showing genus-level interactions among cecal microbes. The different-color nodes represent the phyla of the detected genera. The style of the edge indicates positive (pink solid line) and negative (blue dotted line) correlation, and the line thickness shows the correlation strength. (A) BD, (B) FLA, (C) LFVW, (D) MFVW and (E) HFVW. Only significant correlations are shown (cutoff threshold, Spearman's coefficient > 0.7 and *P* < 0.5).

environmental factors, such as age and diet (40, 41). Previous studies reported that dietary fiber supplementation has no significant effect on the gut microbiota diversity of laying hens during the grower and developer phases (9 to 20 weeks of age) and the finisher phase (89 weeks of age) (42, 43). Furthermore, studies on the gut microbiota in the early life of laying hens are very limited. These results confirmed the conjecture of this study that the starter period is the key period for the development of the gut microbiota in laying hens and that a diet with FVW as a source of fiber could increase the diversity of the gut microbiota during this period. Supplementation of almond hulls (rich in insoluble dietary fiber) had no effect on broiler cecal microbiota diversity (44), while alfalfa (rich in fiber) increased intestinal microbiota diversity (45). The different results may be due to the fact that the fiber sources in FVW are mainly hemicellulose and cellulose. Previous studies have shown that antibiotic growth promoters

**TABLE 1** Effect of dietary supplementation of FVW on SCFA concentration in chickens

| Phase | SCFAs | Concn (mmol/L) in chickens fed diet with[a]: | | | | |
|---|---|---|---|---|---|---|
| | | BD | FLA | LFVW | MFVW | HFVW |
| Prestarter | Acetate | 7.26 ± 0.10 b | 6.62 ± 0.03 a | 6.92 ± 0.11 ab | 9.05 ± 0.44 c | 9.39 ± 0.33 c |
| | Propionate | 0.28 ± 0.01 c | 0.25 ± 0.00 b | 0.32 ± 0.01 d | 0.25 ± 0.00 b | 0.17 ± 0.03 a |
| | Butyrate | 1.08 ± 0.03 e | 1.01 ± 0.04 d | 0.94 ± 0.02 c | 0.89 ± 0.02 b | 0.44 ± 0.00 a |
| | Total SCFAs | 8.62 ± 0.11 b | 7.88 ± 0.01 a | 8.19 ± 0.11 a b | 10.19 ± 0.45 c | 9.99 ± 0.33 c |
| Starter | Acetate | 5.32 ± 0.58 a | 15.24 ± 1.74 c | 10.88 ± 1.29 b | 11.64 ± 1.29 b | 13.16 ± 1.31 bc |
| | Propionate | 0.42 ± 0.01 a | 0.59 ± 0.03 b | 0.37 ± 0.01 a | 1.25 ± 0.05 c | 0.57 ± 0.07 b |
| | Butyrate | 0.57 ± 0.04 a | 1.83 ± 0.07 b | 1.70 ± 0.04 b | 1.74 ± 0.14 b | 1.84 ± 0.05 b |
| | Total SCFAs | 6.31 ± 0.61 a | 17.66 ± 1.74 c | 12.95 ± 1.32 b | 14.62 ± 1.40 b | 15.57 ± 1.74 bc |
| Grower | Acetate | 5.53 ± 0.25 b | 8.21 ± 0.17 d | 6.00 ± 0.21 c | 5.11 ± 0.19 a | 4.78 ± 0.13 a |
| | Propionate | 1.13 ± 0.04 c | 1.06 ± 0.11 c | 0.55 ± 0.03 a | 0.89 ± 0.05 b | 0.53 ± 0.06 a |
| | Butyrate | 0.68 ± 0.04 ab | 0.98 ± 0.04 c | 0.71 ± 0.04 b | 0.61 ± 0.05 a | 0.71 ± 0.03 b |
| | Total SCFAs | 7.34 ± 0.31 c | 10.25 ± 0.15 d | 7.26 ± 0.17 c | 6.61 ± 0.20 b | 6.02 ± 0.15 a |
| Developer | Acetate | 6.24 ± 0.03 d | 7.41 ± 0.08 e | 4.44 ± 0.28 c | 3.78 ± 0.24 b | 2.74 ± 0.06 a |
| | Propionate | 1.51 ± 0.03 c | 1.93 ± 0.14 d | 0.91 ± 0.08 b | 0.75 ± 0.02 a | 0.67 ± 0.07 a |
| | Butyrate | 0.46 ± 0.04 c | 0.57 ± 0.04 d | 0.5 ± 0.16 c | 0.36 ± 0.04 b | 0.28 ± 0.02 a |
| | Total SCFAs | 8.22 ± 0.06 d | 9.92 ± 0.12 e | 5.85 ± 0.22 c | 4.88 ± 0.19 b | 3.69 ± 0.11 a |
| Finisher | Acetate | 4.20 ± 0.45 c | 2.45 ± 0.32 b | 2.83 ± 0.41 b | 2.34 ± 0.18 b | 1.66 ± 0.02 a |
| | Propionate | 1.63 ± 0.06 d | 0.88 ± 0.04 b | 1.06 ± 0.06 c | 0.59 ± 0.04 a | 0.52 ± 0.03 a |
| | Butyrate | 0.71 ± 0.06 d | 0.39 ± 0.01 b | 0.55 ± 0.04 c | 0.34 ± 0.02 b | 0.24 ± 0.01 a |
| | Total SCFAs | 6.54 ± 0.43 d | 3.72 ± 0.34 b | 4.43 ± 0.08 c | 3.27 ± 0.17 b | 2.42 ± 0.05 a |

[a]Values are means and standard deviations. Letters indicate significance ($P < 0.05$).

(flavomycin and virginiamycin) could increase the richness of the gut microbiota in broiler chickens but have no significant effect on the diversity (46), which is consistent with the results of this study showing that the subtherapeutic dose of flavomycin did not destroy the diversity of the gut microbiota. In this study, the structure of the gut microbiota showed convergence and stability with increasing age, which is in agreement with the previous results obtained with laying hens (47). However, the heterogeneity of the gut microbiota structure in broiler chickens increased over time (48). This contrasting result suggests that the farming duration and chicken type lead to differences in the structural development of the gut microbiota. Our results emphasized that the starter phase (0 to 28 days) may the optimal time for FVW to intervene and influence the microbiome.

*Firmicutes* was the absolute dominant phylum during the first 4 weeks of the laying hens' life, in response to the high-protein feed composition during the brood stage of bone and muscle development. *Bacteroidetes* began to become the dominant phylum after the grower phase of laying hens in response to the increase in dietary fiber. The long-term dietary intervention led to a change in gut microbiota intestinal type (31). An increase in the abundance of *Sutterella*, *Synergistes*, *Anaerostipes*, *Ruminiclostridium*, and *Rikenellaceae* in the gut caused by long-term addition of FVW was also observed in this study. *Sutterella* is an obligately anaerobic, Gram-negative bacterium (49), and there is a negative correlation between the presence or abundance of *Sutterella* and the host inflammatory cytokine response (50, 51). *Synergistes* is related to reduce gastrointestinal inflammation and enhances immune function (52). *Anaerostipes* is a butyrate producer, which could convert lactate and acetic acid and sugar to butyrate (53–56). *Ruminiclostridium* is an anaerobic Gram-positive cellulolytic bacterium that produces a variety of carbohydrate-active enzymes (CAZymes) and catabolizes xyloglucan into glucose, xylose, galactose, and cellobiose (57). It includes extracellular multienzyme complexes known as fibrosomes with different specificities to enhance the degradation of cellulosic biomass (58). FLA causes enrichment of *Ruminococcaceae*, *Desulfovibrio*, *Peptococcus*, *Enterorhabdus*, and *Faecalicoccus*. *Ruminococcaceae* is a recognized SCFA-producing bacterium (59).

*Desulfovibrio* is a Gram-negative anaerobe belonging to the sulfate reducing group (60). Sulfate-reducing bacteria could use organic compounds (lactate, propionate, and butyrate) as sources of energy and carbon (61, 62). The expansion of *Desulfovibrio* has been reported to be associated with inflammatory bowel disease, including ulcerative colitis (63–65). *Peptococcus* was found to be strongly positively correlated with the body weight (BW) and average daily gain (ADG) in pig culture experiments (66). In an immune experiment with mice, *Peptococcus* was found to be positively correlated with LPS, D-lactic acid, and TNF-$\alpha$ (67). *Enterorhabdus* is an obesity-promoting bacterium associated with diabetes and other metabolic diseases, while the overgrowth of *Enterorhabdus* is also a sign of ecological imbalance after antibiotic use (68–70). The enrichment of *Faecalicoccus* is usually associated with intestinal inflammation, including ulcerative colitis and Crohn's disease (71). These results are consistent with previous findings that dietary-fiber-rich byproducts could modulate the composition of the cecal microbiota of chickens (72–74).

In addition, FVW increased the positive-relationship cluster between *Firmicutes* and *Bacteroidetes*. Previous studies have shown a strong negative relationship between *Bacteroidales* and *Clostridiales* in the cecum of chickens fed a basic diet (75). *Firmicutes* and *Bacteroidetes* represented most of the anaerobic fermentative bacteria (76), which may present nutritional competition for fermentation substrates in the cecum (77). The gut microbiome typically relies on carbohydrates as its energy source, and the gut microbes that use the same energy source occupy the same niche and form a competitive symbiotic relationship (78, 79). FVW might provide abundant dietary fiber for the gut microbiota, thereby changing the interaction between members of the microbiota (from competitive symbiosis to mutualistic symbiosis). In contrast with previous studies, this study supports the intervention effect of dietary fiber supplementations on the microbe-microbe interaction (19, 80, 81).

Subsequently, FVW increased acetate and propionate, but not butyrate. Intestinal bacteria ferment dietary fiber to produce short-chain fatty acids, which play an important role in immune regulation (82). The result is consistent with the previous report showing that insoluble dietary fiber can increase the content of acetate and propionate in the cecum (83). *Bacteroidetes* could ferment dietary fiber to produce acetate, isovalerate, and succinate, of which succinate is the raw material required for propionate production (84). Acetate strengthens the barrier function by mediating the signaling pathways that enable B cells and goblet cells to secrete mucins and IgA (85). Propionate could promote the development of Treg cells and reduce the expansion of inflammatory Th17 cells (86). Long-term propionate delivery in the colon improved glucose homeostasis, along with the suppression of systemic imperial inflammation (87). These results suggest that FVW may regulate the intestinal immunity by increasing acetate and propionate.

The intestinal immune system must maintain a delicate balance between tolerance of commensal microbiota and immunity to pathogens, maintaining low responsiveness to the commensal microbiota at steady state (88). Previous studies have shown that supplementation with dietary fiber could reduce levels of the proinflammatory factors TNF-$\alpha$, IL-1$\beta$, and IL-6 (89). In this study, FVW decreased the levels of the proinflammatory cytokines TNF-$\alpha$ and IL-6 and increased the levels of the anti-inflammatory cytokine IL-4. Additionally, the biomarker of the intestinal mucosa immune response is the production of secretory immunoglobulin A (S-IgA). It is the most prominent antibody present in mucosal surfaces and protects the intestinal mucosa against the invasion of enteric toxins and pathogenic microorganisms (13, 90). In this study, FVW increased the levels of S-IgA in the small-intestinal mucosa. Similar to our results, dietary fiber (wheat bran and sugar beet pulp) and prebiotics (xylo-oligosaccharides and mannooligosaccharides) increased the amount of S-IgA in the small intestine (91–93). The above results further proved that FVW has the potential of prebiotics to regulate the host immunity microbes in steady state.

In conclusion, we found that FVW, which is rich in dietary fiber, altered the interaction between members of the gut microbiota, regulated the balance between gut

**TABLE 2** Basal composition and nutrient levels of chicken diet

| Ingredient or component | % in diet at phase | | | |
|---|---|---|---|---|
| | **Starter** | **Grower** | **Developer** | **Finisher** |
| Composition | | | | |
| Maize corn | 63.14 | 67.25 | 67.50 | 55.70 |
| Soybean oil | 0.60 | 1.00 | 0.00 | 2.80 |
| Soybean meal | 32.20 | 27.50 | 25.50 | 28.20 |
| Lysine | 0.20 | 0.20 | 0.20 | 0.20 |
| Methionine | 0.25 | 0.20 | 0.25 | 0.25 |
| Dicalcium | 2.20 | 2.30 | 3.00 | 3.60 |
| Limestone | 1.00 | 1.00 | 3.10 | 8.80 |
| Common salt | 0.21 | 0.35 | 0.25 | 0.25 |
| Vitamins and minerals[a] | 0.20 | 0.20 | 0.20 | 0.20 |
| | | | | |
| Calculated nutrition level[b] | | | | |
| DM | 90.65 | 90.50 | 91.34 | 91.12 |
| ME (MJ/kg) | 12.05 | 12.21 | 11.69 | 11.70 |
| CP | 20.02 | 18.25 | 16.87 | 17.00 |
| Ca | 0.99 | 1.00 | 1.11 | 4.11 |
| P | 0.50 | 0.51 | 0.58 | 0.72 |
| EE | 3.41 | 0.39 | 0.29 | 0.52 |
| CF | 2.91 | 2.70 | 2.58 | 2.56 |
| Lys | 1.17 | 1.05 | 1.00 | 1.05 |
| Methionine | 0.53 | 0.46 | 0.50 | 0.50 |
| Cystine | 0.32 | 0.30 | 0.29 | 0.28 |

[a]The premix provided is as follows (per kilogram of feed): vitamin A, 4,500 IU; vitamin D3, 1,200 IU; DL-$\alpha$-tocopheryl acetate, 2,500 IU; vitamin $B_1$, 5,000 mg; vitamin $B_2$, 20,000 mg; vitamin K, 10,000 mg; niacin, 45,000 mg; pantothenic acid, 35,000 mg; biotin, 1,500 mg; folic acid, 3,000 mg; vitamin $B_{12}$, 40 mg; zinc, 45 mg; manganese, 50 mg; iron, 30 mg; copper, 4 mg; cobalt, 100 $\mu$g; iodine, 1 mg; selenium, 100 $\mu$g.

[b]DM, dry matter; ME, metabolizable energy; CP, crude protein; Ca, calcium; P, phosphorus; EE, ether extract; CF, crude fiber.

microbiota and host immunity, and kept the gut microbiota in a healthy and stable state. Against the background of an antibiotic-free culture, this study provides data supporting the development and application of *Flammulina velutipes* stipe wastes as potential prebiotics for laying hens. This study was a small-scale farming experiment under laboratory conditions, which could not fully simulate the factory feeding conditions and environmental stress caused by a large breeding base. Therefore, focusing on the interaction network and the competition and cooperation between members of the gut microbiota will be the next direction of our research.

## MATERIALS AND METHODS

**Experimental design and sample collection.** A 490-day study assessing the impact of *Flammulina velutipes* stipe wastes (FVW) on the cecal microbiota of the laying chicken was performed in Animal Feeding Room, Jilin Agricultural University, Changchun, China. FVW were provided by China Changchun Xuerong Biotechnology Co., Ltd. The collected FVW were naturally dried and then transferred to a feed factory for further use (Jilin Hanhong Animal Husbandry Co., Ltd.). A total of 450 ISA brown laying chicks, purchased from a commercial hatchery, were randomly divided into 5 groups (3 replicates/group, 30 chickens/replicate): BD (basic diet), FLA (basic diet supplemented with 5 ppm flavomycin), LFVW (basic diet supplemented with 2% FVW), MFVW (basic diet supplemented with 4% FVW), and HFVW (basic diet supplemented with 6% FVW). The different groups were fed *ad libitum* with starter feed from 1 to 28 days, grower from 29 to 70 days, developer from 71 to 112 days, and finisher from 113 to 490 days (Fig. 1). The nutritional components of *Flammulina velutipes* stipe wastes are shown in Table S1. All feedings were applied according to the NRC-1994 norms and the principle of equal energy and equal nitrogen (Table 2 and Tables S2 to S5).

The size of the brooding cage was 60 cm by 40 cm by 50 cm (length, width, and height, respectively), and there were 14 chickens per cage from 1 to 56 days and 7 chickens per cage from 57 to 112 days. At 112 days, the laying hens were moved to a laying cage (100 cm by 60 cm by 50 cm [length, width, and height, respectively]), with 3 chickens per cage. No veterinary treatment was required for the duration of the experiment.

Nine chickens per supplementation group were sacrificed at 5 defined time points: day 7 (prestarter), day 28 (starter), day 70 (grower), day 112 (developer), and day 490 (finisher). Intestinal and cecal samples were collected in College of Life Science Building, Jilin Agricultural University. Samples were quickly flash-frozen in liquid nitrogen and stored at −80°C until further processing.

**DNA extraction.** The microbial DNA was extracted from digesta in ceca using the EZNA soil kit (Omega Bio-Tek, USA) according to the manufacturer's protocol. The final DNA concentration and purification were determined with a NanoDrop 2000 instrument, and DNA quality was detected by 1% agarose gel electrophoresis.

**PCR-DGGE and clonal sequencing.** The V3 hypervariable regions of the bacterial 16S rRNA gene were amplified with the primers F338-GC (5′-CGCCCGCCGCGCGCGGGGGGGCGGGGCGGGGGCAGGGGGGCCTCGG AGGCAGCAG-3′) and R518 (5′-ATTACCGCGGCTGCTGG-3′) with a thermocycler PCR system (MG25+; Thermo Scientific, China). The PCRs were conducted using the following program: 95℃ predenaturation for 5 min, 30 cycles (95℃ denaturation for 1 min, annealing at 60℃ for 1 min, and extension at 72℃ for 1 min), with a final extension at 72℃ for 5 min. The PCRs were performed in triplicate using a 25-$\mu$L mixture containing 12.5 $\mu$L premix *Taq* mix, 10 $\mu$L sterilized ultrapure water, 0.5 $\mu$L forward primer, 0.5 $\mu$L reverse primer, and 1.5 $\mu$L of template DNA. A 230-bp DNA fragment was obtained and further analyzed by denaturing gradient gel electrophoresis (DGGE) with a denaturant gradient of 40% to 65% and a concentration of polyacrylamide gel of 8%. The gel was cut to recover clear consensus and specific bands in the DGGE map. The reamplified DNA fragment (without GC clamp) was purified, linked to the pESI-T vector, and transformed into DH5$\alpha$ cells. The positive clones were screened and sequenced at Sangon Biotech (Shanghai, China) Co., Ltd.

**16S rRNA high-throughput sequencing.** The V3-V4 hypervariable regions of the bacterial 16S rRNA gene were amplified with primers F338 (5′-ACTCCTACGGGAGGCAGCAG-3′) and R806 (5′-GGACTA CHVGGGTWTCTAAT-3′) with a thermocycler PCR system (GeneAmp 9700; ABI, USA). The PCRs were conducted using the following program: 95℃ predenaturation for 3 min, 27 cycles of 95℃ denaturation for 30 s, annealing at 55℃ for 30 s, and extension at 72℃ for 30 s, and a final extension at 72℃ for 10 min. The PCRs were performed in triplicate 20-$\mu$L mixtures containing 4 $\mu$L of 5× FastPfu buffer, 2 $\mu$L of a 2.5 mM concentration of deoxynucleoside triphosphates (dNTPs), 0.8 $\mu$L primer (5 $\mu$M), 0.4 $\mu$L FastPfu polymerase, and 10 ng of template DNA. The resulted PCR products were extracted from a 2% agarose gel, further purified by using the AxyPrep DNA gel extraction kit (Axygen Biosciences, USA), and quantified using a QuantiFluor-ST system (Promega, USA) according to the manufacturer's protocol. Purified amplicons were pooled in equimolar amounts and paired-end sequenced (2 × 300 bp) on an Illumina MiSeq platform (Illumina, San Diego, CA, USA) according to the standard protocol by Majorbio Bio-Pharm Technology Co., Ltd. (Shanghai, China).

**Determination of SCFAs concentration.** The experimental conditions were as follows: the chromatographic column was a DB-FFAP capillary column (30 m by 250 $\mu$m by 0.25 $\mu$m), the inlet temperature was 220℃, the flame ionization detector (FID) temperature was 250℃, the column temperature heating program consisted of an initial temperature of 65℃ and then an increase to 190℃ at a 20℃/min heating rate, the split ratio was 25:1, and gas flow rates were 25 mL/min for carrier gas ($N_2$), 40 mL/min for $H_2$, and 400 mL/min for air. The levels of acetate, propionate, and butyrate were detected with an Agilent gas chromatograph 7890A (Agilent Technologies, USA). The standard solution (mixed standard) used was 60 mmol/L (3.60 g/L) of acetate, 50 mmol/L (3.72 g/L) of propionate, and 20 mmol/L (1.76 g/L) of butyrate.

**Determination of cytokines in intestinal mucosa.** TNF-$\alpha$, IL-6, IL-17, IL-2, IL-4, IL-10, and S-IgA in the intestinal mucosa were measured by using chicken-specific enzyme-linked immunosorbent assay (ELISA) quantitation kits (Lengton Bioscience Co. Ltd., Shang Hai, China) according to the instructions of the manufacturer.

**Data analyses.** Quantity One v.4.6.2 was used to analyze the fingerprint of PCR-DGGE and the gray value of each band. For microbial diversity analysis, the richness index is the number of bands, the Shannon-Wiener index was calculated as $-\sum [\ln(ni/\sum ni) \times (ni/\sum ni)] = -\sum [\ln (ni/\sum ni) \times (ni/\sum ni)]$ (where ni is the gray value of the band), and the evenness index was calculated as $H/\ln S$ (where H is the Shannon-Wiener index and S is the richness index). The differences in the microbiota community structures were evaluated by PCA based on Bray-Curtis dissimilarity values and performed with Canoco v.5. BLAST comparison was performed in GenBank at the NCBI website to obtain the corresponding biological classification information for the bands.

For processing of data sequencing, Raw-fastq files were demultiplexed, quality filtered with Trimmomatic, and merged by FLASH with the following criteria. (i) The reads were truncated at any site receiving an average quality score of <20 over a 50-bp sliding window. (ii) By allowing 2-nucleotide mismatches, primers were exactly matched with the removal of possessing ambiguous reads. (ii) Sequences with an overlap of more than 10 bp were mixed according to their overlap sequences. Operational taxonomic units (OTUs) were accumulated with a similarity cutoff value of 97% by using UPARSE (version 7.1 [http://drive5.com/uparse/]); chimeric sequences were elaborated and removed by using UCHIME. The 16S rRNA gene sequence taxonomy was examined by the RDP Classifier algorithm (http://rdp.cme.msu.edu/) against the SILVA 138 16S rRNA database using a confidence threshold of 70%. Networks were then constructed by using the method implemented in Cytoscape v.3.7.1.

All data are expressed as means and standard deviations (SD) as determined by SPSS v.25. The results were analyzed with one-way analysis of variance (ANOVA) and Duncan's multiple-comparison test. The differences were considered statistically significant at a *P* value of <0.05. All box plots, stacked bar charts, and bar charts were drawn using GraphPad Prism v.8.

**Ethical approval.** All procedures in this project were conducted within the ethical regulations and standards set and carried out by the Animal Care Review Committee of Jilin Agricultural University (ID: 2019-08-28-001).

**Data availability.** Sequence data generated in this study have been made available at the Sequence Read Archive (SRA) on NCBI under project number PRJNA628749.

## SUPPLEMENTAL MATERIAL

Supplemental material is available online only.

**FIG S1**, TIF file, 0.9 MB.
**FIG S2**, TIF file, 0.7 MB.
**FIG S3**, TIF file, 1.3 MB.
**TABLE S1**, DOCX file, 0.02 MB.
**TABLE S2**, DOCX file, 0.02 MB.
**TABLE S3**, DOCX file, 0.02 MB.
**TABLE S4**, DOCX file, 0.02 MB.
**TABLE S5**, DOCX file, 0.02 MB.

## ACKNOWLEDGMENTS

We express sincere gratitude to the doctoral students, especially Shad Mahfuz and Shuyuan Wang, who provided unconditional help during the experiment.

We have no conflict of interest to declare.

This work was supported by Department of Science and Technology of Jilin Province (no. JJKH20190917KJ).

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
