## [Reviewer comments · mSystems]

Longitudinal Study on the Effects of *Flammulina velutipes* Stipe Wastes on Cecal Microbiota of Laying Hens

Jiali Wei, Huanwei Xiao, Yingbo Wei, Ivan St`eve Nguepi Tsopmejio, Chang Sun, Haoyuan Wu, Zhouyu Jin, and Hui Song

Corresponding Author(s): Hui Song, Jilin Agricultural University

Review Timeline:

Submission Date:	September 2, 2022
Editorial Decision:	October 5, 2022
Revision Received:	November 16, 2022
Accepted:	November 20, 2022

Editor: Suzanne Ishaq

Reviewer(s): Disclosure of reviewer identity is with reference to reviewer comments included in decision letter(s). The following individuals involved in review of your submission have agreed to reveal their identity: Yunhe Xu (Reviewer #1)

Transaction Report:

DOI: <https://doi.org/10.1128/msystems.00835-22>

October 5, 2022

Prof. Hui Song
Jilin Agricultural University
Chanchun
China

Re: mSystems00835-22 (Longitudinal Study on the Effects of *Flammulina velutipes* Stipe Wastes on Cecal Microbiota of Laying Hens)

Dear Prof. Hui Song:

Thank you for submitting your manuscript to mSystems. We have completed our review and I am pleased to inform you that, in principle, we expect to accept it for publication in mSystems. However, acceptance will not be final until you have adequately addressed the reviewer comments.

Preparing Revision Guidelines

Sincerely,

Suzanne Ishaq

Editor, mSystems

Journals Department
American Society for Microbiology
1752 N St., NW

Reviewer comments:

Reviewer #1 (Comments for the Author):

1. In the experimental design, L303, 9 replicas/group, and Figure 1 are inconsistent. In L719, Table 1, in the formula table, the nutritional component indicators CP, Ca, P, EE, etc. are usually expressed in %.
2. As for diversity index, it would be more illustrative if PCR-DGGE results could be compared with high-throughput sequencing results.
3. The statistical analysis methods used in the article are rich and comprehensive, the charts are beautiful and standardized. The discussion is related to the reasoning, but the comparison through the available publications in the literature are limited.
4. All data were expressed as mean {plus minus} standard deviation (SD) in statistical analysis, while SEM was used in the results (Table 2)
5. The author should provide the advantages of PCR-DGGE compared to the types of high-throughput sequencing.
6. A clearer illustration of contribution should be further provided in the conclusion.

Reviewer #2 (Comments for the Author):

In general, this is a valuable study, which made a comprehensive study on the influence of dietary fiber on gut microbiota of laying hens throughout the cycle. However, some sentences were too long. Authors are advised to reorganize long sentences to improve readability. Here are some specific suggestions:

Line 42-46: "SCFAs can also act as signaling molecules to regulate host immunity. It is the case of the combination of propionate and butyrate that effectively inhibited LPS-induced inflammatory response of Treg cells and reduced the production of inflammatory cytokines such as IL-6 and IL-12 (8)." LPS should have its full name when it first appears in the manuscript.

Line 80-83: "Laying hens living up to approximately 70 weeks, before their laying rate decreased to about 65%, while previous studies have reported that microbial changes at different ages could influence outcome and prevent interstudy comparisons." This sentence is too long. It can be split into two sentences. In addition, "while previous studies have reported that microbial changes at different ages could influence outcome and prevent interstudy comparisons" need references to prove it.

Line 83-87: "In this study, we followed neonatal laying hens from hatching to 70 weeks to estimate the effects of FVW or flavomycin on the cecal microbiota by cloning sequencing and 16SrRNA gene analysis. We here describe the changes in dominant bacterial taxa in the gut microbiota as a result of FVW or flavomycin supplementation at different feeding stages." This sentence is best used in the passive voice. It is better to add the purpose or significance of this study after this sentence.

Line 92-94: "Ninety chickens were randomly divided into 3 replicates in each group, and received 5 ppm flavomycin, LFVW (2%FVW), MFVW (4%FVW), HFVW (6%FVW), or BD (no supplementation)." Flavomycin should be flavomycin (FLV).

Line 109-110: "Thus, the increase in community diversity was accompanied by the decreased heterogeneity of the cecal microbiota." Concluding language should be placed at the end of the paragraph.

Line 165-167: "The microbiota in the cecum ferments undigested carbohydrates by using their own glycohydrolytic activity to produce SCFAs. SCFAs play a role in regulating immunity by acting on epithelial and immune cells." This sentence need references to prove it.

Line 198-199: "Globally, FVW could thus regulate the dynamic balance of intestinal mucosal immunity in laying hens and ensured a steady-state level of commensal microbiota."

This sentence would be better to be changed as follows:

"Therefore, FVW could regulate the dynamic balance of intestinal mucosal immunity in laying hens, which might be beneficial to the homeostasis level of commensal microbiota."

Table 1 and 2: The number of digits reserved after the decimal point in the table should be consistent.

Table 1: Some punctuation is missing in the footnotes. Dietary formulations and nutrient levels for the other groups should be provided, and the nutrient composition of the FVP should also be listed. These can be shown in the supplementation tables.

Longitudinal Study on the Effects of Flammulina velutipes Stipe Wastes on Cecal Microbiota of Laying Hens

In general, this is a valuable study, which made a comprehensive study on the influence of dietary fiber on gut microbiota of laying hens throughout the cycle. However, some sentences were too long. Authors are advised to reorganize long sentences to improve readability. Here are some specific suggestions:

Line 42-46: “SCFAs can also act as signaling molecules to regulate host immunity. It is the case of the combination of propionate and butyrate that effectively inhibited LPS-induced inflammatory response of Treg cells and reduced the production of inflammatory cytokines such as IL-6 and IL-12 (8).” **LPS should have its full name when it first appears in the manuscript.**

Line 80-83: “Laying hens living up to approximately 70 weeks, before their laying rate decreased to about 65%, while previous studies have reported that microbial changes at different ages could influence outcome and prevent interstudy comparisons.” **This sentence is too long. It can be split into two sentences. In addition, “while previous studies have reported that microbial changes at different ages could influence outcome and prevent interstudy comparisons” need references to prove it.**

Line 83-87: “In this study, we followed neonatal laying hens from hatching to 70 weeks to estimate the effects of FVW or flavomycin on the cecal microbiota by cloning sequencing and 16SrRNA gene analysis. We here describe the changes in dominant bacterial taxa in the gut microbiota as a result of FVW or flavomycin supplementation at different feeding stages.” **This sentence is best used in the passive voice. It is better to add the purpose or significance of this study after this sentence.**

Line 92-94: “Ninety chickens were randomly divided into 3 replicates in each group, and received 5 ppm flavomycin, LFVW (2%FVW), MFVW (4%FVW), HFVW (6%FVW), or BD (no supplementation).” **Flavomycin should be flavomycin (FLV).**

Line 109-110: “Thus, the increase in community diversity was accompanied by the decreased heterogeneity of the cecal microbiota.” **Concluding language should be placed at the end of the paragraph.**

Line 165-167: “The microbiota in the cecum ferments undigested carbohydrates by using their own glycohydrolytic activity to produce SCFAs. SCFAs play a role in regulating immunity by acting on epithelial and immune cells.” **This sentence need references to prove it.**

Line 198-199: “Globally, FVW could thus regulate the dynamic balance of intestinal mucosal immunity in laying hens and ensured a steady-state level of commensal microbiota.”

This sentence would be better to be changed as follows:

“Therefore, FVW could regulate the dynamic balance of intestinal mucosal immunity in laying hens, which might be beneficial to the homeostasis level of commensal microbiota.”

Table 1 and 2: The number of digits reserved after the decimal point in the table should be consistent.

Table 1: Some punctuation is missing in the footnotes. Dietary formulations and nutrient levels for the other groups should be provided, and the nutrient composition of the FVP should also be listed. These can be shown in the supplementation tables.

Response to Reviewers

Dear Editor-in-Chief,

Thank you for giving us the opportunity to submit a revised draft of our manuscript entitled “Longitudinal Study on the Effects of *Flammulina velutipes* Stipe Wastes on Cecal Microbiota of Laying Hens” by Jiali Wei *et al.* submitted to your prestigious journal *mSystems*. We strongly appreciated the reviewers for their precious time in reviewing our paper and providing valuable comments, questions and suggestions. They were valuable and insightful comments that led us the possibility to improve the current version of our manuscript. We have carefully considered any comments, suggestions and questions to try our best to address every one of them. We hope the manuscript, after careful revisions, will meet your high standard. The authors welcome further constructive comments if any. We provided below the point-by-point responses.

Sincerely,

Song Hui,

School of Life Science,

Jilin Agricultural University,

130118 Changchun, P. R. China.

Phone: +86-13604449943; songhuisk@jlau.edu.cn.

Reviewer #1 (Comments for the Author):

1. *In the experimental design, L303, 9 replicas/group, and Figure 1 are inconsistent. In L719, Table 1, in the formula table, the nutritional component indicators CP, Ca, P, EE, etc. are usually expressed in %.*

Response: Thank you for this requirement. The grouping described in L303 was corrected to conform with Figure 1. Nutritional component indicators in L719 were expressed as percentages (%).

2. *As for diversity index, it would be more illustrative if PCR-DGGE results could be compared with high-throughput sequencing results.*

Response: Thank you for this suggestion. Because the sampling time points of clone sequencing (PCR-DGGE) were day 7 (prestarter phase), day 28 (starter phase), day 70 (grower phase), day 112 (developer phase) and day 490 (finisher phase), while the sampling time points of high-throughput sequencing were day 490 (finisher phase), the diversity indices of the two sequencing methods were compared at finisher phase. There was no statistically significant difference in the alpha and beta diversity indices, which is shown in Supplementary Fig S1.

3. *The statistical analysis methods used in the article are rich and comprehensive, the charts are beautiful and standardized. The discussion is related to the reasoning, but the comparison through the available publications in the literature are limited.*

Response:: Thank you for your comment. Comparisons with other available publications were enriched.

4. *All data were expressed as mean {plus minus} standard deviation (SD) in statistical analysis, while SEM was used in the results (Table 2)*

Response: Thank you for this remind. The data in Table 2 was modified to data \pm standard deviation (SD).

5. *The author should provide the advantages of PCR-DGGE compared to the types of high-throughput sequencing.*

Response: Thanks for this suggestion. Due to the two sequencing methods were not performed in parallel, DGGE-clone sequencing was not compared with high-throughput sequencing in this study. These two sequencing methods have their own advantages and serve different research requires. The DGGE-clone sequencing method has the advantages of low cost, good repeatability and low data analysis intensity. It can show the composition and distribution of dominant microbiota, which is suitable for exploring the changes of dominant microbiota in cecum of laying hens at different breeding phases in this study. High-throughput sequencing can detect the changes of microbiota with small abundance, which not only proves the correctness of DGGE-clone sequencing results, but also comprehensively and specifically proves the enrichment of differential microbiota in cecum and the changes of interaction network between intestinal microbiota of laying hens fed *Flammulina velutipes* stip wastes for a long time.

6. *A clearer illustration of contribution should be further provided in the conclusion.*

Response: Thank you for your advice. The contribution of this work was added in L294-296.

Reviewer #2 (Comments for the Author):

In general, this is a valuable study, which made a comprehensive study on the influence of dietary fiber on gut microbiota of laying hens throughout the cycle. However, some sentences were too long. Authors are advised to reorganize long sentences to improve readability. Here are some specific suggestions:

Response: Thank you very much for your appreciation.

1. *Line 42-46: "SCFAs can also act as signaling molecules to regulate host immunity. It is the case of the combination of propionate and butyrate that effectively inhibited LPS-induced inflammatory response of Treg cells and reduced the production of inflammatory cytokines such as IL-6 and IL-12 (8)." LPS should have its full name when it first appears in the manuscript.*

Response: Thank you for this remind. The full name of LPS was added in Line 44.

2. *Line 80-83: "Laying hens living up to approximately 70 weeks, before their laying rate decreased to about 65%, while previous studies have reported that microbial changes at different ages could influence outcome and prevent interstudy comparisons." This sentence is too long. It can be split into two sentences. In addition, "while previous studies have reported that microbial changes at different ages could influence outcome and prevent interstudy comparisons" need references to prove it.*

Response: Thank you for this remark. The long sentence in line 80-83 was split into two sentences. The references were added in Line 83.

3. *Line 83-87: "In this study, we followed neonatal laying hens from hatching to 70 weeks to estimate the effects of FVW or flavomycin on the cecal microbiota by cloning sequencing and 16SrRNA gene analysis. We here describe the changes in*

dominant bacterial taxa in the gut microbiota as a result of FVW or flavomycin supplementation at different feeding stages.” This sentence is best used in the passive voice. It is better to add the purpose or significance of this study after this sentence.

Response: Thanks for this suggestion. The sentence in line 83-87 was revised as the passive voice. And the purpose was added in Line 87-90.

4. *Line 92-94: “Ninety chickens were randomly divided into 3 replicates in each group, and received 5 ppm flavomycin, LFVW (2%FVW), MFVW (4%FVW), HFVW (6%FVW), or BD (no supplementation).” Flavomycin should be flavomycin (FLV).*

Response: Thanks for this remind. The abbreviation of flavomycin has been modified to FLA in Line 96, which is consistent with the figure in the article.

5. *Line 109-110: “Thus, the increase in community diversity was accompanied by the decreased heterogeneity of the cecal microbiota.” Concluding language should be placed at the end of the paragraph.*

Response: Thank you for this remark. The sentence was placed at the end of the paragraph in Line 120-121.

6. *Line 165-167: “The microbiota in the cecum ferments undigested carbohydrates by using their own glycohydrolytic activity to produce SCFAs. SCFAs play a role in regulating immunity by acting on epithelial and immune cells.” This sentence need references to prove it.*

Response: Thank you for this remind. The references were added in Line 169-170.

7. *Line 198-199: “Globally, FVW could thus regulate the dynamic balance of intestinal mucosal immunity in laying hens and ensured a steady-state level of commensal microbiota.”*

This sentence would be better to be changed as follows: “Therefore, FVW could

regulate the dynamic balance of intestinal mucosal immunity in laying hens, which might be beneficial to the homeostasis level of commensal microbiota.”

Response: Thanks for your advice. The sentence was changed in Line 201-202 as suggested.

8. *Table 1 and 2: The number of digits reserved after the decimal point in the table should be consistent.*

Response: Thank you for this requirement. The numbers in Tables 1 and 2 were retained the same decimal point.

9. *Table 1: Some punctuation is missing in the footnotes. Dietary formulations and nutrient levels for the other groups should be provided, and the nutrient composition of the FVW should also be listed. These can be shown in the supplementation tables.*

Response: Thanks for this suggestion. The nutrient composition of *Flammulina velutipes* stip wastes and the nutrient levels of each group at each stage were shown in Supplementary Tables S1-S5, respectively.

November 20, 2022

Prof. Hui Song
Jilin Agricultural University
Changchun
China

Re: mSystems00835-22R1 (Longitudinal Study on the Effects of *Flammulina velutipes* Stipe Wastes on Cecal Microbiota of Laying Hens)

Dear Prof. Hui Song:

Your manuscript has been accepted, and I am forwarding it to the ASM Journals Department for publication. For your reference, ASM Journals' address is given below. Before it can be scheduled for publication, your manuscript will be checked by the mSystems production staff to make sure that all elements meet the technical requirements for publication. They will contact you if anything needs to be revised before copyediting and production can begin. Otherwise, you will be notified when your proofs are ready to be viewed.

Publication Fees:

If you would like to submit a potential Featured Image, please email a file and a short legend to mSystems@asmusa.org. Please note that we can only consider images that (i) the authors created or own and (ii) have not been previously published. By submitting, you agree that the image can be used under the same terms as the published article. File requirements: square dimensions (4" x 4"), 300 dpi resolution, RGB colorspace, TIF file format.

We recognize that the video files can become quite large, and so to avoid quality loss ASM suggests sending the video file via <https://www.wetransfer.com/>. When you have a final version of the video and the still ready to share, please send it to mSystems staff at mSystems@asmusa.org.

Sincerely,

Suzanne Ishaq
Editor, mSystems

Journals Department
Supplemental Material: Accept
Table S1: Accept
Tables S4: Accept
Supplemental Material: Accept
Tables S5: Accept
Tables S3: Accept
Table S2: Accept
Supplemental Material: Accept